# Potential Application of Pulsed Field Ablation in Ventricular Arrhythmias

**DOI:** 10.3390/medicina59040723

**Published:** 2023-04-07

**Authors:** Jie Qiu, Meiyan Dai, Yang Bai, Guangzhi Chen

**Affiliations:** Division of Cardiology, Department of Internal Medicine, Tongji Hospital, Tongji Medical College, Huazhong University of Science and Technology, 1095 Jiefang Ave., Wuhan 430030, China

**Keywords:** pulsed field ablation, myocardium, ventricular arrhythmias, coronary venous system, Purkinje system

## Abstract

Pulsed field ablation (PFA) is a new ablative method for the therapy of arrhythmia. Recent preclinical and clinical studies have already demonstrated the feasibility and safety of PFA for the treatment of atrial fibrillation (AF). However, the application of PFA may not be limited to the above fields. There are some data on the application of PFA on ventricular arrhythmias (VAs), such as ventricular fibrillation (VF) and ventricular tachycardia (VT). Further, a case report about PFA has been published recently, in which PFA was successfully applied to the ablation of premature ventricular contractions (PVCs) from the right ventricular outflow tract. Thus, we aimed to review recent research findings of PFA in ventricular ablation and evaluate the possibility of its application in VAs.

## 1. Introduction

Pulsed field ablation (PFA) is a new ablative method for the therapy of arrhythmia. It generates non-thermal, high-energy and short-duration electrical impulses to create lesions by means of electroporation. Within milliseconds, PFA can create sufficiently deep myocardial lesions without damage to non-myocardial collateral tissue, such as pulmonary veins, coronary arteries, the esophagus and nerves. Recent preclinical and clinical studies have already demonstrated the feasibility and safety of PFA for the treatment of atrial fibrillation [1,2]. However, there are fewer data on the application of PFA for ventricular arrhythmias (VAs), such as ventricular tachycardia (VT) and ventricular fibrillation (VF). Surprisingly, a case report about the application of PFA on VAs has been published recently, in which premature ventricular contractions (PVCs), originating from the right ventricular outflow tract, have been successfully obviated [3]. Thus, the aim of our work was to review recent research findings of PFA in ventricular ablation and evaluate the possibility of its application in VAs.

## 2. Effect of PFA on Ventricular Myocardium

Both the epicardial and endocardial pathway of PFA application to ventricular ablation have been reported [4,5,6,7]. Early in 2012, Wittkampf FH et al. [4] performed PFA on the ventricular epicardium in the epicardial swine model with a 7F circular ablation catheter. During this procedure, all PFA applications (50, 100 and 200 J) were delivered on the left ventricular epicardium in different sequences. They found that when PFA was delivered with 200 J, successive deep circular lesions (median depth of 5.2 ± 1.2 mm) were created, in spite of the use of separate electrodes. Additionally, no signs of thermal damage were observed in the ablation site and electrodes directly after PFA. Histological results indicated that cardiomyocytes were completely replaced by granulation tissue, which consists of fibroblasts with capillaries and loose collagen fibers. Similar results were shown in the work of Neven K et al. [5]. In their report, PFA was performed with different energy settings (50, 100 and 200 J) using a 12-mm circular catheter, which was introduced after pericardial access through subxiphoid puncture in the porcine model. In the cross sections of 100- and 200-J settings, transmural myocardial lesions have been observed, and the average depths of the 100- and 200-J lesions were 7.0 ± 2.0 and 11.9 ± 1.5 mm, respectively; the average widths were 16.2 ± 4.3 and 19.8 ± 1.8 mm, respectively. Meanwhile, no obvious intimal hyperplasia of the coronary arteries were observed after 3-month follow-up.

The application of PFA on ventricular myocardium in endocardial procedures has also been proved feasible. In the in vivo study by Koruth JS et al. [6], they demonstrated the feasibility and safety of focal PFA application on ventricular endocardium. In their work, PFA applications were delivered to the right and left ventricles of healthy swine under general anesthesia with a fixed voltage amplitude of 2200 V, using a four-spline multielectrode PFA catheter. Gross measurements showed that average lesion dimension was 22.6 ±4.1 mm wide by 6.5 ± 1.7 mm deep, and the maximum width and depth were 28.6 mm and 9.4 mm, respectively. In addition, authors also found that cardiomyocytes were homogeneously replaced by fibrous tissue with a region of surrounding myocytolysis and no overlying thrombus in PFA lesions, while vasculature and nerve fascicles within surrounding fibrosis were preserved. Similar results have also been shown in a subsequent article [8], which applied PFA energy settings of 1500 V for 120 pulses in the interventricular septum (IVS). Even four weeks after ablation, authors also observed transmural scars formation at the ventricular level, and the width and depth were 6.37 ± 0.62 mm and 5.99 ± 0.75 mm, respectively. Further, one study found that PFA in ventricular myocardium showed repetition dependency using a lattice catheter [9]. In this report, the depth and volume of acute and chronic lesion increased significantly in the group that received four repetitive applications, when compared with the group with single application (*p* < 0.001 for both comparisons). In addition, histological results showed that a well demarcated necrotic core without coagulation necrosis was observed in acute lesions, while chronic lesions showed tissue thinning with fibro-fatty replacement.

## 3. Two Sides of PFA: Pro- and Anti- Arrhythmic Effects

The key mechanism of PFA is the application of high-energy and short-duration electric fields to induce irreversible electroporation (IRE). However, the effect of PFA on cardiomyocytes may also be transient; pores may be resealable and the cells could preserve the vitality, which is called reversible electroporation (RE). After the delivery of PFA, the magnitude of electric fields around the electrodes decreased from the electrodes outward into the tissue. The nature of these fields is such that immediately near the electrodes there is a region in which they cause IRE. Meanwhile, these regions were surrounded by other regions with lower electric fields, in which RE was induced. In both regions of IRE and RE, the enhanced cell membrane permeability may open a channel for ion transport, which may trigger various arrhythmias [10]. One study has investigated the possibility of developing arrhythmias resulting from the enhanced cell permeability induced by electrical pulses but found no dysrhythmias on the ECG [11].

There are other explanations about the two sides of PFA on arrhythmia. Electroporation has been found to selectively affect the bundles of the conduction system and small trabeculated structures, resulting in temporary suppressing of excitability and a conduction block [12]. In the heart of a rabbit model, Vladimir P et al. observed that electrical conduction in a small papillary muscle can be transiently inhibited by a strong electric shock, and such a transient block of electrical conduction can last from one beat to a few seconds, which depended on the strength of electric shocks [13]. Thus, authors suggested that such transient inhibition of the cardiac conduction system may lead to the initiation of a re-entrant or focal arrhythmia. On the other hand, it may also have an anti-arrhythmic effect through isolating the ectopic focus and reducing tissue mass available for the maintenance of arrhythmia.

Since conflicting data indicate the two sides of electroporation on arrhythmia, both pro- and anti-arrhythmic effects, the exact mechanism involved is still not clear. But there is a consensus that delivering IRE within the absolute refractory period of the cardiac cycle would reduce the risk of arrhythmias [2,14]. When electrical stimulation exceeds the threshold excitation potential, it can induce localized depolarization to build into an action potential. Therefore, IRE could trigger a premature action potential in a myocardial cell, leading to the occurrence of arrhythmias, especially VT. Thus, in order to avoid arrhythmias, it was necessary to adjust IRE delivery to fall into the absolute refractory period. Based on the present research, despite the finding that a proarrhythmic effect could be expected of PFA, new arrhythmias were not yet demonstrated.

## 4. Parameters of PFA Procedure during Ventricular Ablation

The optimal PFA parameters for ventricular ablation remain undefined at this nascent stage of development. First, according to current research, monophasic waveform can create the stronger damaging effects on myocardium, while increasing the risk of gas formation and muscle contraction [15,16]; biphasic waveforms can greatly reduce muscle contraction but will generate weaker damaging effects. Second, microsecond PFA delivery will cause obvious skeletal muscle stimulation, and intravenous paralytic should be administered during the process, in order to improve catheter stability. Nanosecond PFA delivery did not result in obvious extracardiac muscle stimulation in the absence of a paralytic, which indicates the priority for PFA on ventricular ablation. Further, fine contact force (CF) is critical to achieve transmural lesion for radiofrequency ablation (RFA), but it may be not necessary for PFA. The main reason for this may be that PFA does not require direct contact with the destruction of the cardiac conduction system and myocardium. Most studies suggest that there is no correlation between CF and lesion depth or volume during PFA. However, some studies found that lesion depth may increase significantly with increasing CF, and the depth increases from 1 to 10 g of CF, then plateaus up to 30 g [17]. However, the potential implications of parameters on the PFA effect have not been explicitly evaluated and needs further investigation (Figure 1).

## 5. PFA and Purkinje-Related Ventricular Arrhythmias

There is mounting evidence that the Purkinje system plays a critical role in both the initiation and perpetuation of VF and VT [18,19]. Organized in an arborized architecture, the Purkinje system consists of specialized cells, which are responsible for the ventricular synchronous activation. Electrophysiologically, its action potentials have specific features, such as a prominent phase 1 upstroke, a long-lasting plateau phase and a more negative diastolic potential. Thus, Purkinje fibers are prone to abnormal automaticity and triggered activity. Previous studies have reported that the Purkinje-related VAs range from isolated ectopies to VT and VF and may occur in patients with or without structural heart disease [19]. The effect of medical therapy is limited in these patients, and RFA has been regarded as an effective way to reduce the need for anti-arrhythmic drugs (AADs) and the number of implantable cardioverter defibrillator (ICD) shocks in this population [20,21]. However, current application of RFA for the Purkinje-related VAs have several limitations, particularly because it can only be delivered in a point-to-point manner, which is time-consuming and tedious.

PFA can successfully injure Purkinje fibers, both reversibly and irreversibly, which may be dose-dependent. Livia C et al. performed electroporation on Purkinje tissues in isolated hearts of a canine model [22]. In this study, direct current (DC) was delivered in a unipolar manner with different dosages from 750 to 2500 V, in 10 pulses with a 90-μs duration at a frequency of 1 Hz. According to their results, all Purkinje potentials were consistently eradicated by IRE at different voltage settings (750, 1000, 1500, 2000 or 2500 V). The Purkinje potential (PP) was initially abolished, but it would recur after 5 min at the lowest energy deliveries of 750 and 1000 V. Furthermore, the potential of the left bundle could not be abolished until a second delivery at 2000 V was applied. Further, the potential of the His bundle could not be abolished at any energy delivery, even with the setting of 2500 V. Thus, based on the above results, we may suppose that IRE eradicates PP without significant injury to His and left bundles within a specific range of voltage settings (perhaps 1500–2000 V). However, this article has only shown the short-term effect of IRE on the Purkinje fibers.

Sugrue A. et al. further evaluate the effect of PFA therapy on Purkinje fibers in the in-vivo acute and chronic canine models [23]. They delivered 10 pulses at a frequency of 0.83–1 Hz with different voltages (750–3000 V for acute models, 500–1500 V for chronic models) and a pulse duration (20 μs for acute models, 90 μs for chronic models). When the pulse duration was set at 90 μs, the PP was persistently lost with a voltage greater than 1050 V; when the pulse duration was set at 20 μs, a threshold of greater than 1800 V was required to record persistent acute loss of the PP. Thus, both voltage and pulse duration are the determining factors for persistent effect of PFA on Purkinje fibers.

Interestingly, both studies have proved that PFA caused little or no damage to the underlying myocardium, which indicating its tissue selectivity. The histopathological analysis has shown minimal myocardial damage after PFA therapy, both in the acute and chronic canine models, manifested as acute endocardial hemorrhage and minimal contraction band necrosis. Although acute damage of Purkinje fibers was characterized by an increase in eosinophilic fibers, which had altered structure and loss of cytoplasmic detail, there was no suggestion of Purkinje damage in the chronic canine model. The His bundle was resistant to the PFA delivered with multiple deliveries, and in the cases of the anterior fascicle, the signal may be lost after increasing doses of PFA, but it then recurs after 5 min.

There was a limit study to evaluate the effect of PFA on VAs via acting on Purkinje tissue. Livia C et al. found that half of the canine models failed to induce VF due to the delivery of electroporation energy to the Purkinje fibers, and there was a significant reduction observed in the window of VF vulnerability (5.7 ± 2.9 J, *p* = 0.0003) [22]. Authors suggested that abolishing Purkinje signals using PFA may be related to an enhanced resistance to VF induction. They even give their hypothesis that PFA may provide a means for delivery of a vector into the Purkinje fibers, providing an opportunity to develop a vaccine for Purkinje-related arrhythmias management.

## 6. PFA and Scar-Related Ventricular Arrhythmias

Scar-related VT is a serious and potentially life-threatening arrhythmia, usually caused by anatomically macro-reentrant circuits that form after a myocardial infarction (MI) or in other types of diseased myocardium [24]. The prevention of scar-related VT relies on ICD, AADs and, more recently, on RFA [25]. Nevertheless, these methods have their own risk of complications and adverse events, with a recurrence rate up to 50% at two years. AADs have limited benefit for ICD shock reduction and increased survival, and it is also associated with severe side effects [26].

ICDs actually do not prevent VT; approximately 10% of patients experience an electrical storm. Furthermore, ICD shocks are often painful and reduce quality of life. Some clinical studies have shown the superiority of RFA over conventional medical therapy in controlling recurrent VT [27,28]; however, the overall complexity of this procedure has precluded its widespread use, and the long-term outcome is not always favorable [29]. In addition, RFA procedure usually takes a longer time, because of the fact that, in order to cause multiple transmural lesions throughout the myocardial scar, the ablation duration of each lesion may be up to 1 min.

Consisting of scattered viable myocardial fibers, collagen fibers and adipose tissue, arrhythmogenic myocardial scar tissue has a complex three-dimensional structure [30]. These scattered viable myocardial fibers are key components of scar-related re-entry and ablation targets of PFA. One study demonstrated that higher resistivity and insulate viable myocardium may be seen in infarcted myocardium from RFA induced injury, with only 10% of RFA lesions even identifiable [31]. Im SI et al. compared the lesion characteristics of PFA and RFA in healthy and MI swine models [32]. Using two different catheters, multispline 8-pole catheter or linear quadripolar, bipolar and biphasic PFA was delivered for 2.5 s × 4 applications in this work. They found that lesion depths of PFA were not greater than RFA energy applied on normal myocardium, while the depth of PFA was markedly greater than that of RFA in MI myocardium (*p* = 0.005). In myocardial scar tissue, the depth of lesion had no significant differences between two PFA catheters (*p* = 0.235). Furthermore, histology also showed that ablation of viable myocardium islands within a scar >10 mm away from the tip of PFA catheter was also observed. The reason may be that the same scar resistivity that insulates a scar from RFA channels the pulse electric field towards myocardial cells, enhancing the ablation of viable myocardium. Thus, PFA may effectively ablate viable myocardial islands within and around scar tissue, and it can be hopefully used to treat scar-related VT.

## 7. PFA and Ventricular Arrhythmias from the Coronary Venous System

Idiopathic VAs have presented as monomorphic non-sustained VT or as frequent PVCs, and they are the most common VAs in patients without structural heart disease [33]. Idiopathic epicardial VAs, originating from the vicinity of the coronary sinus system (CVS), account for about 9% of idiopathic VAs [34]. It is well known that most idiopathic VAs can be eliminated by endocardium RFA, whereas for the patients whose left ventricular outflow tract epicardial origin sharing a perivascular origin, RFA may be not successful. This kind of VA is usually located at the transitional area from the great cardiac vein (GCV) to the anterior interventricular vein (AIV) [35]. Since most idiopathic epicardial VAs originate from perivascular and are usually adjacent to a coronary vein, CVS provides a safer, more convenient and minimally invasive access to epicardial regions of the left ventricle, especially to the transition area between the GCV and AIV. Although previous case reports and series have demonstrated that epicardial RFA through the CVS and its branches is safe and effective, we still need to be aware of the possible occurrence of complications due to thermal or mechanical injury, such as coronary vein rupture or perforation, venous thrombosis and neighboring coronary arteries stenosis [36].

Studies have shown that PFA could create transmural epicardial lesions and deep continuous circular lesions in the pulmonary veins [37,38]. Moreover, animal data indicated that PFA may be a safe ablation procedure for the important nearby tissues, such as the coronary arteries, the esophagus and the phrenic nerve [39,40]. Buist TJ et al. investigated the feasibility of IRE in the coronary sinus in a porcine model [41]. In their report, IRE ablations were performed with 100 J pulses with a modified 9-French steerable linear hexapolar ablation catheter. There were no complications and signs of structural damage of coronary sinus during the procedure and 3-week survival. At half-hour post ablation, 100% isolation was achieved in all animals. At three weeks follow-up, pacing thresholds were significant higher than that of the baseline. Histological analysis also showed transmural ablation lesions in muscular sleeves surrounding the CS. However, authors did not evaluate the effect of different energy settings on lesion size, for the reason that it is difficult to build the relationship between exact location of the energy setting and the corresponding lesion at histological assessment. Thus, more experimental and clinical data are needed to assess the safety and efficacy.

## 8. PFA and Ventricular Arrhythmias from the Interventricular Septum

The interventricular septum (IVS) has been proven to be an important part of the VT substrate in nonischemic cardiomyopathy (NICM). This specific type of VT is uncommon, and its rate of successful ablation is unsatisfactory when using conventional thermal energies such as RFA [42]. One major limitation is that the penetration capability of RFA is limited in thicker myocardial tissue. Even at higher-power settings, usually greater than 40 W, its maximal penetration depth is about 5–6 mm, while the thickness of IVS ranges from 9 to 12 mm. Further, the damage risk to the conducting system and coronary artery in this area during PFA is also a problem. Knowing its ability to produce adequate myocardial lesion while sparing other critical tissues, PFA may also be an attractive way to treat arrhythmia originating from deep IVS. Previous reports have observed similar myocardial lesion depths after PFA with bipolar fashion between catheter electrodes, and the depth of lesions can reach up to 9 mm [8]. However, some argue that although bipolar PFA could produce deep lesions, this is done at the expense of a substantial risk of steam pops [43].

In the work of van Zyl M et al., PFA was applied between identical non-irrigated deflectable ablation catheters positioned on either side of the IVS in chronic canine models [44]. During the procedure, a stable pulse width (100 microseconds or 300 nanoseconds) was delivered with variation in the power settings (1000–1500 V) and pulse number per site (40–60 pulses). According to the imaging and histology results, they confirmed that myocardial ablation was successfully obtained with bipolar PFA delivery across the IVS. A maximum lesion depth of over 10 mm was present with a PFA lesion, which was almost transmural. Conduction abnormalities, such as right bundle branch block, left bundle branch block, fascicular block and complete atrioventricular block, were seen immediately after PFA delivery, but most of them eventually disappeared. Histologically, there are still viable conduction system tissues and Purkinje fibers in most of the ablated myocardium. In addition, this study showed no ECG evidence of acute myocardial ischemia, and widely patent epicardial coronary arteries were proved using coronary angiography. Histologically, small intramural coronary vessels within ventricular PFA lesions were preserved. In conclusion, bipolar PFA for VAs from the IVS is feasible and can produce near transmural lesions.

## 9. Conclusions

Although RFA has become the standard energy source in current ablation procedures for VAs, PFA can provide advantages in certain clinical scenarios. Tissue specificity of PFA has the potential to simultaneously increase safety and efficacy by creating transmural lesions while sparing non-myocardial tissue at risk for injury with RFA. PFA has been already tested clinically in humans for ablation of AF, and this technology also has exciting potential application in VT and VF ablation [45,46] (Table 1).

## Figures and Tables

**Figure 1 medicina-59-00723-f001:**
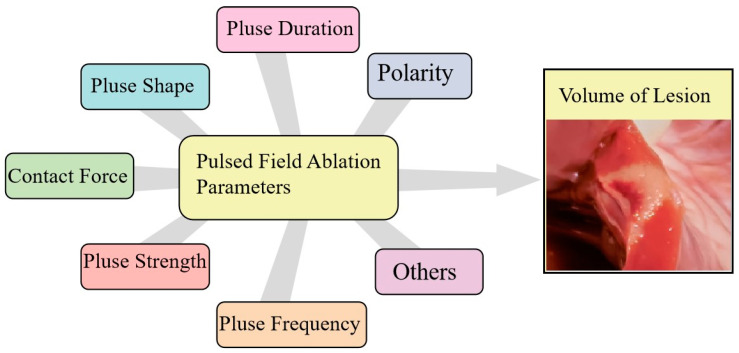
Multiple parameters affect the volume of lesion caused by pulsed field ablation.

**Table 1 medicina-59-00723-t001:** Summary of case reports of pulsed field ablation in ventricular arrhythmias.

Case Report (Author, Year)	Sample Size	Gender	Age	Indications	Comorbid Conditions	Catheter Type	PFA Parameters	Complication
Schmidt B, 2022 [3]	1	Female	48	PVC from originate from RVOT	None	Farawave, Farapulse, Boston Scientific, USA	Pulses of 2.5 s duration with a voltage of 1.8 kV	None
Krause U, 2023 [45]	1	Male	33	Monomorphic VT	Ebstein’s anomaly	Farawave, Farapulse, Boston Scientific, USA	NA	None
Ouss A, 2023 [46]	1	Male	69	Postinfarction VT	Anterior myocardial infarction	Farawave, Farapulse, Boston Scientific, USA	2 kV performed with the biphasic waveform	None

Abbreviations: PFA, pulsed field ablation; PVC, premature ventricular contractions; RVOT, right ventricular outflow tract; VT, ventricular tachycardia.

## Data Availability

There is no new data created in our study.

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
