# Peer review of "Potential Application of Pulsed Field Ablation in Ventricular Arrhythmias"

_medicina, 2023, doi:10.3390/medicina59040723_

Round 1

Reviewer 1 Report

Dear Sirs,

I read with great interest the manustricpt entitled: „ Potential Application of Pulsed Field Ablation in Ventricular 2 Arrhythmias.” PFA is currently a hot topic in the field of invasive arrhythmia management.

The submitted paper systematises current knowledge on the ablation of ventricular arrhythmias with PFA. There are currently quite a few publications on the use of PFA on an animal model. Experience with ablation in humans using PFA is increasing. However, there currently appears to be no standard regarding the curvature of the electrode, the number of applications and the energy to be used to achieve a sustained effective therapeutic effect. This is evidenced by the number of citations in this area in the submitted paper. The paper highlights the potential benefits and most common risks associated with the use of PFA. It does not focus on the technical details of the PFA during a single procedure.

The authors analysed the current publications, paying particular attention to the effects of PFA on the ventricular muscle, the efficacy in VT therapy and the possible limitations of this method.

Knowledge in this area is still incomplete and numerous studies are planned to address this issue. In my opinion, supplementing the manuscript with the technical aspect of PFA ablation.... i.e. the methodology of a single procedure ..., e.g. where applicate the energy ... on critical isthmus VT, whether during arrhythmia or based on substrate mapping .... supported by possible publications in this area - would enrich the publication and make it more interesting for the readers.

Papers concerning PFA is currently a hot topic.... and knowledge on it is sought after.

Yours sincerely

Reviewer

Author Response

Dear Reviewer: 

Thank you for your comments concerning our manuscript entitled “Potential Application of Pulsed Field Ablation in Ventricular Arrhythmias” . Those comments are valuable and very helpful for revising and improving our paper. We have studied comments carefully and have made correction which we hope meet with approval. The main corrections in the paper and the responds to the reviewer’s comments are as following: 

Responds to the reviewer’s comments: 

Response to comment:  “Knowledge in this area is still incomplete and numerous studies are planned to address this issue. ... supported by possible publications in this area - would enrich the publication and make it more interesting for the readers.”

Response: The optimal PFA parameters for ventricular ablation remain undefined. We have added related content about parameters of PFA on ventricular ablation, and cited the required references (mark in red).

 We appreciate for Editors/Reviewers’ warm work, and hope that the correction will meet with approval. Once again, thank you very much for you comments and suggestions.

   Sincerely yours,

Guangzhi Chen.

Reviewer 2 Report

The paper by Qiu et al focuses on a very interesting issue: the possibility to deliver pulse field ablation in ventricle and its possible benefits in the treatment of ventricular arrhythmias.

The paper is well written and understandable.

Ventricular arrhythmias are interesting divided in different groups according to the underlying cause. 

The studies are well described and clinical implication are clear.

The main issue is the recent pubblication of a group of studies not included in the citations (i.e. Kawamura I, Reddy VY, Wang BJ, Dukkipati SR, Chaudhry HW, Santos-Gallego CG, Koruth JS. Pulsed Field Ablation of the Porcine Ventricle Using a Focal Lattice-Tip Catheter. Circ Arrhythm Electrophysiol. 2022 Sep;15(9):e011120. doi: 10.1161/CIRCEP.122.011120. Epub 2022 Sep 8. PMID: 36074657; PMCID: PMC9794124.)

In my opinion, the first reference is inappropriate. It is a self citation of a paper about the same argument.

Minor issues

In the paragraph about "Two sides of PFA: pro- and anti- arrhythmic effects", I suggest to underline that, despite a proarrhythmic effect could be expected, new arrhythmias were not demonstrated. 

Author Response

Dear Reviewer: 

Thank you for your comments concerning our manuscript entitled “Potential Application of Pulsed Field Ablation in Ventricular Arrhythmias” . Those comments are valuable and very helpful for revising and improving our paper. We have studied comments carefully and have made correction which we hope meet with approval. The main corrections in the paper and the responds to the reviewer’s comments are as following: 

Responds to the reviewer’s comments: 

1.Response to comment:  “The main issue is the recent pubblication of a group of studies not included in the citations (i.e. Kawamura I, Reddy VY, Wang BJ, Dukkipati SR, Chaudhry HW, Santos-Gallego CG, Koruth JS. Pulsed Field Ablation of the Porcine Ventricle Using a Focal Lattice-Tip Catheter. Circ Arrhythm Electrophysiol. 2022 Sep;15(9):e011120. doi: 10.1161/CIRCEP.122.011120. Epub 2022 Sep 8. PMID: 36074657; PMCID: PMC9794124.)”

Response: Required reference has been added.

2.Response to comment:  “In my opinion, the first reference is inappropriate. It is a self citation of a paper about the same argument.”

Response: The first reference has been deleted.

3.Response to comment:  “Two sides of PFA: pro- and anti- arrhythmic effects", I suggest to underline that, despite a proarrhythmic effect could be expected, new arrhythmias were not demonstrated.”

Response: Related statement has been added in the section "Two sides of PFA: pro- and anti- arrhythmic effects"(mark in red).

 We appreciate for Editors/Reviewers’ warm work, and hope that the correction will meet with approval. Once again, thank you very much for you comments and suggestions.

   Sincerely yours,

Guangzhi Chen.
